# Key Properties of a Bioactive Ag-SiO_2_/TiO_2_ Coating on NiTi Shape Memory Alloy as Necessary at the Development of a New Class of Biomedical Materials

**DOI:** 10.3390/ijms22020507

**Published:** 2021-01-06

**Authors:** Mateusz Dulski, Robert Gawecki, Sławomir Sułowicz, Michal Cichomski, Alicja Kazek-Kęsik, Marta Wala, Katarzyna Leśniak-Ziółkowska, Wojciech Simka, Anna Mrozek-Wilczkiewicz, Magdalena Gawęda, Maciej Sitarz, Karolina Dudek

**Affiliations:** 1Institute of Materials Engineering, Faculty of Computer Science and Materials Science and Silesian Center for Education and Interdisciplinary Research, University of Silesia in Katowice, 75 Pulku Piechoty 1A, 41-500 Chorzow, Poland; 2A. Chełkowski Institute of Physics, Faculty of Computer Science and Materials Science and Silesian Center for Education and Interdisciplinary Research, University of Silesia in Katowice, 75 Pulku Piechoty 1A, 41-500 Chorzow, Poland; robert.gawecki@us.edu.pl (R.G.); anna.mrozek-wilczkiewicz@us.edu.pl (A.M.-W.); 3Institute of Biology, Biotechnology and Environmental Protection, Faculty of Natural Sciences, University of Silesia in Katowice, Jagiellonska 28, 40-032 Katowice, Poland; slawomir.sulowicz@us.edu.pl; 4Department of Materials Technology and Chemistry, Faculty of Chemistry, University of Lodz, Pomorska 163, 90-236 Lodz, Poland; michal.cichomski@chemia.uni.lodz.pl; 5Faculty of Chemistry, Silesian University of Technology, B. Krzywoustego 6, 44-100 Gliwice, Poland; alicja.kazek-kesik@polsl.pl (A.K.-K.); marta.wala@polsl.pl (M.W.); katarzyna.lesniak-ziolkowska@polsl.pl (K.L.-Z.); wojciech.simka@polsl.pl (W.S.); 6Faculty of Materials Science & Ceramics, AGH University of Science and Technology, Mickiewicza 30, 30-059 Cracow, Poland; mgaweda@agh.edu.pl (M.G.); msitarz@agh.edu.pl (M.S.); 7Refractory Materials Division in Gliwice, Łukasiewicz Research Network—Institute of Ceramics and Building Materials, Toszecka 99, 44-100 Gliwice, Poland

**Keywords:** hybrid Ag-SiO_2_ coating, wettability, tribology, surface roughness, ion release, antimicrobial studies, cell viability, cell adhesion

## Abstract

Recent years have seen the dynamic development of methods for functionalizing the surface of implants using biomaterials that can mimic the physical and mechanical nature of native tissue, prevent the formation of bacterial biofilm, promote osteoconduction, and have the ability to sustain cell proliferation. One of the concepts for achieving this goal, which is presented in this work, is to functionalize the surface of NiTi shape memory alloy by an atypical glass-like nanocomposite that consists of SiO_2_-TiO_2_ with silver nanoparticles. However, determining the potential medical uses of bio(nano)coating prepared in this way requires an analysis of its surface roughness, tribology, or wettability, especially in the context of the commonly used reference coat-forming hydroxyapatite (HAp). According to our results, the surface roughness ranged between (112 ± 3) nm (Ag-SiO_2_)—(141 ± 5) nm (HAp), the water contact angle was in the range (74.8 ± 1.6)° (Ag-SiO_2_)—(70.6 ± 1.2)° (HAp), while the surface free energy was in the range of 45.4 mJ/m^2^ (Ag-SiO_2_)—46.8 mJ/m^2^ (HAp). The adhesive force and friction coefficient were determined to be 1.04 (Ag-SiO_2_)—1.14 (HAp) and 0.247 ± 0.012 (Ag-SiO_2_) and 0.397 ± 0.034 (HAp), respectively. The chemical data showed that the release of the metal, mainly Ni from the covered NiTi substrate or Ag from Ag-SiO_2_ coating had a negligible effect. It was revealed that the NiTi alloy that was coated with Ag-SiO_2_ did not favor the formation of *E. coli* or *S. aureus* biofilm compared to the HAp-coated alloy. Moreover, both approaches to surface functionalization indicated good viability of the normal human dermal fibroblast and osteoblast cells and confirmed the high osteoconductive features of the biomaterial. The similarities of both types of coat-forming materials indicate an excellent potential of the silver-silica composite as a new material for the functionalization of the surface of a biomaterial and the development of a new type of functionalized implants.

## 1. Introduction

Nickel-titanium alloys (NiTi), which have a near-equiatomic chemical composition, have the superelasticity and shape memory effects that are associated with a reversible phase transformation. These features result in NiTi alloys being widely used in many biomedical applications [1,2]. This is also associated with their acceptable biocompatibility (comparable to stainless steel or titanium and titanium-based implant materials) [3,4,5], the high strength, relatively low stiffness, high toughness, and their shape-recovery behavior, which ensures its good mechanical stability within the host tissue. As a result, NiTi alloys are commonly used as maxillofacial and dental implants, lumbar vertebral replacements, joint replacements, bone plates, bone tissue engineering, spine fracture fixation, anchorage, and repair [6,7,8,9,10,11]. Unfortunately, their long-term use in vivo revealed some disadvantages of NiTi alloys, which causes the release of toxic nickel ions into the body, and limits the full acceptance of an implant by the organism [12,13,14]. This is one of the most problematic issues that has to be resolved in order to develop a durable implant material.

In the literature, many various methods and materials have been reported that can be used to functionalize and improve the applicability of NiTi for long-term applications [15,16,17,18,19,20,21]. However, it should be stressed that the coatings that are used to functionalize NiTi alloys cannot be too thick and/or rigid because this can possibly limit the shape changes that are associated with the shape memory effects. Thus, one of the methods that has been widely developed recently, which enables layers of a controlled thickness to be produced, is electrophoretic deposition (EPD). This approach is widely used in engineering complex hybrid composite layers on substrates with irregular shapes and/or morphologies [17,22,23,24,25,26,27,28]. Unfortunately, the ceramic coatings that are fabricated using EPD require high-temperature treatment to improve the adhesion of the deposited material to a metallic substrate. As a result, another limitation associated with the thermal NiTi phase stability should be emphasized. Here, high-temperature annealing usually leads to the decomposition of the B2 parent phase into equilibrium (Ti_2_Ni and/or Ni_3_Ti) and/or non-equilibrium (Ni_4_Ti_3_) phases [29], which may negatively impact the shape memory effects and superelasticity [30,31].

Some of the most common materials that are used in regenerative medicine are calcium phosphates (CaPs), i.e., hydroxylapatite, calcium triphosphate, and their modified forms. These materials are highly biocompatible with bone tissue, have a high level of corrosion resistance, compressive strength, and hardness. They are also considered to be materials that enhance cell proliferation and cell adhesion. Unfortunately, the mechanical properties of calcium phosphates, including their low fracture toughness, low formability when they come into contact with tissues, low strength or low Weibull modulus result in their limited applications, mostly as an addition to induce the bioactivity of the implants when they come into contact with bone tissue [32,33,34,35,36]. Another problem of calcium phosphates in application is the fact that coatings that are prepared using such ceramics are prone to develop bacterial biofilms [37,38,39], as well as to delaminate or crack after high-temperature sintering. As a result, implant materials that are coated with calcium phosphate ceramics can be susceptible to bacteria adhesion, and their colonization on the surface. Thus, biomaterials that are prepared in this way are subject to bacterial infections or favor the formation of infectious reactions. If the CaPs coatings are porous, harmful ions can migrate from the metallic substrate through them and then promote the development of allergies. Hence, this kind of final material cannot be applicable to an organism in the long term.

Hence, one of the completely new routes for improving the surface of a NiTi alloy using a composite nanostructure composed of silica matrix with silver nanoparticles dispersed homogeneously within the matrix has recently appeared [40]. Hypothetically, this solution should enable the formation of stable coatings that gradually release metal nanoparticles from the matrix, as well as improving the biocompatibility and bioactivity of the final material. It is worth noting that silver also has antibacterial properties, but unfortunately, is also prone to the release of ions into the environment, which may lead to the death of human cells. Additionally, at high concentrations, it stimulates or even enhances its toxicity effect against the biological systems. All of the negative aspects are associated with the high reactivity of silver in the biological environment, which usually translates into its blocking the respiratory chain at the cytochromes [41,42], inducing a massive proton leakage through the bacterial outer membrane, or inhibiting DNA replication [43]. On the other hand, improving the coat-forming material by applying silver nanoparticles at a concentration lower than 2 at.% prevents microbial biofilm formation, improves the pathogenic features of the coat-forming material, and stimulates human cell proliferation [44,45,46,47]. Silica and its compounds, in turn, are nontoxic, inert and according to many literature studies reduce long-term wear, prevent corrosion, limit the dissolution effects, and increase the stability of the ionic forms of metals (e.g., Ag, Cu) [47,48,49]. All of these features are the reasons that silica glass and/or silica-based composites such as polydimethylsiloxane (PDMS) are commonly used in orthopedics and other branches of medicine [50,51,52], including in the regeneration process (e.g., after fractures), and the treatment of bone defects in the hip, knee, and spine [53]. Unfortunately, sintering the material may cause structural and surface modifications that can change the physicochemical features in an unexpected way. Some literature studies highlighted that sintering the silver-silica coating at temperatures as high as 800 °C led to the formation of a continuous complex coating with physicochemical features that were different to those for the non-heat-treated material [54,55]. In turn, it should be mentioned that calcination at 450 °C is usually considered to be enough to achieve a balance between the thermal stability of a system and the antimicrobial features of a material. It was determined that biomaterials that had been functionalized in this way still had the advantages that are associated with the antimicrobial effect, which was even more visible for other common materials after the thermal-treatment procedure, e.g., calcium phosphates [56,57,58]. Interestingly, studies dealing with the problems associated with temperature in the context of silver-silica-based coatings usually focus on steel or titanium substrates [56,57,58], and there are not many papers that consider the problem of NiTi alloy functionalization with these kinds of materials. Thus, calcium phosphates remain the main target material for functionalizing the NiTi alloys, while the hybrid biocomposites that are built that are based on them are simultaneously being developed as a new class of coat-forming biomaterials [17,29,59,60,61]. Additionally, there is little scientific data regarding the silver-silica-based coatings and the problem of calcination, and its impact on physicochemical features of the functionalized biomaterial [40]. Moreover, the EPD approach when considered in the context of the codeposition of such materials has not yet been used. Additionally, only the first two papers considered the problem of surface functionalization using silver-silica nanomaterials that have recently been developed by Dudek et al. [40] and Dulski et al. [62] in the context of functionalizing the NiTi surface. Such an approach was determined to be important for improving the stability of the coated substrate, especially in the context of long-term in vivo use and the biological response.

It is also worth mentioning that preparing a new type of coating not only requires structural and physicochemical studies, but also an analysis of the surface roughness, mechanical properties, bioactivity, wettability, cell biocompatibility, and antibacterial properties [63]. Those data are extremely important after the functionalized biomaterials have been sintered in higher thermal conditions (~450 °C) or even at very high temperatures as in the case of a NiTi alloy (700–800 °C). To respond to these issues, a series of literature studies showed that surface roughness or wettability should be considered in the context of the biological response [64]. Here, the significant increase in the surface roughness of more than 0.2 mm and/or of the surface-free energy facilitated the formation of unwanted bacterial biofilm [65,66]. However, the material with a surface roughness that ranged between 10–100 nm did not affect the total amount of human cell adhesion [67,68]. Another paper illustrated a negative correlation between the surface roughness and water contact angle (WCA), while the improvement of cell adhesion, growth, and viability was associated with the very low rough surfaces of up to 30 nm and high hydrophilicity [69]. In turn, surface roughness that was estimated as close to 70 nm for ceramic coatings implied an enhancement of the cell adhesion parameters [70]. It is worth noting that those data are usually obtained by more local atomic force microscopy (AFM) studies, while slightly different results may result from the more global profilometry studies. Another crucial factor of the new types of coatings is determining the stability of the biomaterial in the organism system [71,72,73] in which implants usually interact with the environment of body fluids. They are usually implicated in or stimulate the corrosion processes, and the same ions are released from the surface of the implant and accumulate in the surrounding tissues [74]. Titanium is considered to be a biologically neutral element that may cause allergies only at higher concentrations [75]. In turn, nickel or silver, especially in elevated content belong to the toxic, carcinogenic, and allergic metals [76,77]. Thus, functionalizing the surface of a NiTi alloy by different types of materials, including metals and composites is a key factor for increasing its biocompatibility.

Hence, functionalizing the biomedical NiTi alloy by using a new class of biologically active silver-silica composite and characterizing the surface topography, roughness, wetting ability, and friction coefficient properties using atomic force microscopy (AFM) and water contact angle (WCA) was thoroughly investigated. Finally, those parameters were correlated with metal ion release, bioactivity in a simulated body fluid (SBF) solution, and the antimicrobial response, cell viability, and cell adhesion, primarily in the context of osseointegration. All of the results indicated the biologically friendly nature of the composite that may indicate its potential usefulness as a new class of functionalized implant material, which could lead to the faster development of tissue engineering, implantology and regenerative medicine.

## 2. Results and Discussion

The deposition and post-heat-treatment parameters of a silver-silica coating on the surface of a NiTi alloy and its impact on the morphology and structure were previously studied using X-ray diffraction (XRD), Raman spectroscopy, X-ray photoelectron spectroscopy (XPS), and scanning electron microscopy (SEM) that was equipped with an energy dispersive spectrometer (EDS) [62]. The crack-free coatings that homogeneously covered the entire surface of a NiTi alloy were engineered using deposition parameters (5 V, 15 min) and post-heat treatment (700 °C/2 h). In this solution, the coat-forming material was composed of a nanometer silver-silica layer with irregular silica agglomerates embedded within this layer. Structurally, the pristine material, which was composed of silver carbonate, coexisted with the metallic and metal-organic silver complexes that were spread within the nonstoichiometric silica. The heat treatment caused the coalescence of the ceramic particles and a structural reorganization of the deposited material. As a result of the structural analysis, it was determined that the temperature led to the formation of a complex structure consisting of SiO_2_-TiO_2_ glass with silver that stabilized the network of such a structure. At this point, it is worth considering the potential use of such a system in the context of the further suitability in vitro or even in vivo. For this purpose, the combination of the microstructure and physicochemical parameters that accompany the newly formed composite coatings should be combined. Other important factors that also have to be considered from the applicational point of view are the surface parameters such as roughness, wettability, or the friction coefficient. These are sensitive to the surface modification methods and should be optimized when fabricating the material. Additionally, selecting the appropriate parameters and analyzing any changes due to mechanical or thermal treatment should also be verified. Therefore, we thoroughly investigated many features of the coatings before and after sintering as well as for the reference hydroxyapatite (HAp). All of these features are crucial for understanding the behavior of the coat-forming material and enabling the most prospective coatings for biological studies to be prepared.

The morphology and the microstructure at the micro- and nanoscale, which were investigated using atomic force microscopy (AFM), were very similar to each other for the Ag-SiO_2_ coating before and after sintering, with spherical and/or ellipsoidal objects that were spread homogenously across the surface (Figure 1a,c). These data are also consistent with the results from the SEM investigations and were comparable to the Raman images that were obtained for nonthermally treated and thermally modified samples [62]. Greater differences appeared only at a high magnification, which at the microscale showed large objects composed of agglomerated tiny and small particles that were closely adjacent to each other, while the borders between them were blurred and difficult to clearly distinguish after sintering (Figure 1b,d). Moreover, to look at the morphological features of the deposited material more precisely, the roughness values from two different scales for five images that were recorded for different ((40 × 40) μm and (2 × 2) μm) areas were averaged. According to this analysis, the root mean square (RMS) parameters averaged over several images of the pristine sample were equal to (139 ± 2) nm for (40 × 40) μm and (25.9 ± 0.8) nm for (2 × 2) μm, respectively. When compared to the RMS values of the uncoated NiTi surface of (2.6 ± 2) nm, this data clearly illustrates a relatively rough surface for the silver-silica coating [29]. It is interesting that after sintering, at the microscale the RMS parameter decreased slightly to (112 ± 3) nm considering (40 × 40) μm area, while at the nanoscale it increased almost two-fold to (39.5 ± 0.5) nm in (2 × 2) μm area. While this data indicates slight global surface morphology modifications that were much higher on the local scale, the RMS values still indicate the rough features of the coating. The data for the reference hydroxyapatite coat-forming material was also quite similar, which suggests the biomedical usefulness of the silver-silica material. In turn, those values were much higher than the RMS values that were previously in the literature for coatings composed of calcium phosphates [29]. However, it seems that surface roughness is acceptable, especially in the context of (i) the free access of the collagen molecules from the interstitial fluid, (ii) the formation of natural hydroxyapatite, (iii) the stimulation of cell proliferation or (iv) cell adhesion. However, surface roughness is only one of the parameters that shapes the surface biological properties of implants and cannot be considered individually.

Hence, other factors such as the wettability parameters including the water contact angle (WCA), the surface free energy (SFE), and tribological features should also be discussed in more detail in the context of developing a new composite (Table 1). More precisely, wettability is a crucial parameter that favors the osteoinduction process and influences the absorption of molecules, thereby promoting the adhesion of fibroblasts and/or bacteria [78]. Hydrophilic surfaces should also have a better biological activity when they come into contact with body fluids; provide higher osseointegration [79,80] and due to the permeation of the culture liquid in the structures, affect the better proliferation of fibroblast or osteoblast cells. In turn, determining the mechanical features is extremely important in the context of medical devices including implants, which usually come into contact with the native tissues, which causes various effects, e.g., frictional forces. Thus, a biological material with inappropriate tribological parameters may lead to discomfort or heat due to friction.

According to theoretical assumptions, the silver-silica coatings should have a hydrophilic character as well as the tribological features that are similar to the reference calcium phosphates. As a result of the wettability study, the water contact angle for the pristine material was close to (65.7 ± 2.1)° and increased slightly to (74.8 ± 1.6)° after sintering (Table 1). The slightly higher contact angle values for the reference biocompatible hydroxyapatite resulted from the chemical features of the silica, which is usually added as a modifier to produce a more hydrophobic surface. Another explanation for this effect may be the formation of a more porous structure inside the silica with air trapped inside it. Another interesting correlation is the similar trend in the roughness and wettability that was observed for Ag-SiO_2_ and hydroxyapatite when they were compared. Here, there were higher WCA values for the surface with slightly higher roughness parameters, probably the result of a structural reorganization and the incorporation of the carbon compounds (e.g., amorphous carbon) into the coat-forming material as was described based on the XPS and Raman data [62]. In this context, amorphous carbon might be responsible for the formation of more-ordered layers with fewer defects compared to the nonsintered coating. However, both coatings still had hydrophilic parameters despite the increase in the contact angle (Ѳ) for the heat-treated sample.

The relationship between the water contact angle and the individual components of the surface free energy is estimated based on the theoretical Young−Dupre equation [81]. According to this, the surface free energy of the deposited Ag-SiO_2_ coating was estimated as (61.1 ± 0.5) mJ/m^2^, which was comparable to the data for hydroxyapatite coating (Table 1), but both values were significantly lower relative to the literature data [82]. The slight discrepancies between the literature and the experimental data confirmed the formation of water monolayer films and proved the hydrophilic properties of the surface. In turn, the slightly lower polar and higher dispersive component compared to the reference and Ag-SiO_2_ coatings might be due to the carbon dioxide adsorption on the surface of the silver nanoparticles and the siloxy units with the CH_x_ (x = 2,3) moieties [83]. The chemical modification that occurred within the Ag-SiO_2_ coat-forming material after sintering implied a strong decrease in the SFE polar values, while the dispersive parameter was almost unchanged. Such alterations corresponded with the chemical and structural data that was discussed based on the XPS and Raman data, which was primarily due to the rearrangement of the silica network, the decomposition of the siloxy units, and the evaporation of the adsorbed C-O groups [62]. In turn, the incorporation of amorphous carbon into the structure of the new SiO_2_-TiO_2_ composite preserves the nonpolar character of the coating. Moreover, the parameters were only slightly lower relative to the reference hydroxyapatite, which confirmed the high potential of the studied coat-forming material.

The tribological studies of the deposited silver-silica composite coating indicated a relatively high friction coefficient value f = 0.4, which corresponded well with the value that was obtained for the hydroxyapatite coating (Table 1). In turn, the slightly higher friction coefficient compared to the uncoated reference NiTi material [84] may have been caused by the presence of adsorbed water molecules on the surface of the silver nanoparticles, silica matrix (Ag-SiO_2_), or calcium phosphates (e.g., HAp). What is more, it seems that the considered coat-forming materials facilitated the tribochemical reaction at the tribological interface, which was in good agreement with the previously discussed WCA and SFE values. At the same time, it indicated the necessity of the surface functionalization of biomedical implants. The impact of temperature on the tribological features of the composite surface caused a decrease in the friction coefficient value to the level of the nonfunctionalized NiTi for the Ag-SiO_2_ composite. This effect may correspond to the decreased hydrophilicity that was found during the wettability measurements and may have resulted from the formation of additional organic layers that prevented the formation of capillary films. In this context, the reorganization of the Ag-SiO_2_ surface and incorporation of the amorphous carbon into the structure is not without significance. Despite this fact, the tribological parameters remained at an acceptable level, which enables the Ag-SiO_2_ layer to be considered as a potential new material for functionalizing NiTi alloys and further in vivo experiments.

According to the preliminary studies, all of the parameters previously considered did not change after sintering relative to the reference and Ag-SiO_2_ materials. Thus, silver-silica coatings can be considered to be prospective coat-forming materials for the future functionalization of NiTi alloy. It should be mentioned that the surface and chemical parameters could also have an impact on the biological features, primarily in the context of the microbial biofilm formation, cytotoxicity, and cell adhesion. Therefore, the following results are discussed only in the context of the sintered material.

There was no antimicrobial inhibition of microbial growth on the reference coating that was composed of hydroxyapatite (Figure 2a,b). Moreover this coating even led to the formation of the Gram-negative (*E. coli*) bacterial biofilm that overgrows the edge of NiTi alloy plates (Figure 2a). At the same time, the hydroxyapatite coating did not promote biofilm formation on the edge of the alloy by the Gram-positive bacteria (*S. aureus*) (Figure 2b). The negative correlation between the results is quite interesting and is difficult to explain unambiguously, especially in the context of the previous studies that are discussed in the paper. Because our data correlated with the literature, we might have expected that the presence of phosphates and/or the hydroxyl groups anchoring on the hydroxyapatite surface might facilitate the formation of a water film on the surface. This might have favored the overgrowing of Gram-negative *E. coli*, which is the first phase of biofilm formation, and is known due to its motility [85]. As for the lack of the *S. aureus* biofilm formation in our study, a similar effect was previously observed for a titanium substrate [86]. This effect might be correlated with the hydrophilic character of the HAp coatings (estimated to be between 65° and 75°) and the surface roughness (close to 100 nm and lower), which limited or even inhibited the bacterial growth [64,82]. Unfortunately, the physicochemical features were not sufficient to protect the material from *E. coli* colonization. Hence, the different behaviors of the hydroxyapatite coatings relative to the various bacteria strains make it challenging to draw general conclusions. As a result, we decided to modify the approach correlated with the hydroxyapatite coat-forming material with a silver-silica composite with very similar wettability and roughness parameters (Table 1) and investigate its bacterial response. Data obtained for silver-silica-coated NiTi alloy revealed a lack of the *E. coli* biofilm that overgrows the edge of NiTi alloy plates (Figure 2c) as well as the appearance of a slight inhibition zone, which was observed after 28 days. A similar effect was observed for the composite material that had been exposed to *S. aureus* (Figure 2d), for which the inhibition zone still occurred near the edge of the silver-silica-covered NiTi alloy after a few hours of contact of the material with the bacteria. As a result, the mechanism that is responsible for inhibiting the biofilm formation could be associated with a synergistic interaction between the SFE, roughness, and probably also the chemical composition. The fact that the presence of the phosphate or hydroxyl groups was not crucial for protecting the coating as was suggested in some literature reports because of the similar or even better antimicrobial effect that was found for the silver-silica material was also interesting. However, the negligible release rate of silver and the low release rate of silicon or nickel requires another explanation. Here, it was necessary to look at the surface charge or the chemical environment. The XPS data revealed that the surface of the coat-forming material was composed of nonstoichiometric silica and hydroxyl groups that were anchored to the surface of the silica, which was probably negatively charged as well as the oxidized and metallic silver that limited the bacteria from coming into contact with the coating [62]. Similar results were also obtained for the coatings that had been prepared as a combination of silver-hydroxyapatite [87], silver-hydroxyapatite polymer coatings [86], or β-TCP + Ag-SiO_2_ [44].

The ion concentration after immersion in the Ringer solution revealed that Ni ions were released on the uncoated sample and the Ag-SiO_2_ functionalized NiTi alloy (Table 2 and Table 3). The Ni ions were released from the substrate and increased for both samples over time. The higher concentration of Ni ions that were detected for the uncoated samples of the silver-silica coating composite indicated the formation of a protective barrier. Furthermore, the concentration of the Ni ions for the uncoated material significantly increased after 16 days of the immersion of the sample to (1.24 ± 0.77) mg/L, while the coated material remained more or less on the same level. The Ti and Ag ions that were released from both samples were in a concentration below the detection limit (<0.01 mg/L). This observation may correlate with the formation of SiO_2_-TiO_2_ composite, which was previously described [62]. The low release of the silver ions, which could be explained by the silver ions being trapped within the SiO_2_-TiO_2_ layer, and it being immobilized in the interstitial positions was also interesting. In turn, the analysis of the Ringer solution of the Ag-SiO_2_-functionalized NiTi alloy showed that a low amount of Si ions had been released from the coating. More precisely, after 24 h of the immersion of the sample, the concentration of the Si ions was (0.03 ± 0.01) mg/L and increased to (0.08 ± 0.01) mg/L after the next four days of incubation. When the sample was immersed in the Ringer solution, the concentration of Si ions increased and after 23 days, it was (0.20 ± 0.01) mg/L. According to this data, Si ions seem to be partially associated with the SiO_2_-TiO_2_ interlayer as was shown by the XPS and Raman data as well as with the agglomerates, which were probably more unstable and prone to release [62]. The Ca and P ions that were detected during the experiment were caused by the Ringer solution.

SEM images of the silver-silica functionalized material after a three-week immersion in an SBF showed the crystallization of agglomerated spheroid globules that covered the surface homogenously (Figure 3). The stoichiometric ratio of Ca/P, which was estimated as being equal to 1.68, is comparable to the literature data for the hydroxyapatite and hydroxyapatite-based coatings [88,89] and higher than the estimate for the uncoated NiTi surface with the stoichiometric ratio Ca/P estimated, which was equal to 1.44. As a result, like the hydroxyapatite substrate, the silver-silica coating stimulated the formation of the typical hydroxyapatite, which confirmed its bioactive properties.

Lastly, the cell viability results that were obtained for two normal human cell lines, i.e., human normal fibroblasts (NHDF) and human osteoblasts (HOB) after 72 h indicated that there was no statistically significant decrease in cell viability compared to the NiTi alloy that had been functionalized by the hydroxyapatite and Ag-SiO_2_ coatings (Figure 4a). This observation is in good agreement with the physicochemical data that were previously described and illustrates the similar parameters of both coatings. Despite the lack of statistically significant differences between the cell lines, there was a decrease in cell viability for the NHDF (86.8%) and HOB (73.6%) cell lines that had been grown on the Ag-SiO_2_ coating compared to the hydroxyapatite. This behavior can be explained by assuming the chemical or morphological aspect. From the chemical point of view, an important factor might be associated with the release of metal ions into the medium, wherein only the presence of nickel, titanium, or silver ions may cause lower cell viability (Table 2 and Table 3). Interestingly, the effect of silver and titanium seemed to be negligible, while the other elements such as Ca or P probably biostimulate and favor the cell viability. Thus, the presence of nickel ions might be associated with the decrease in cell viability, wherein the number of ions that were released into the solution was still low and therefore its impact on the cell viability was insignificant. This is especially visible when our data is compared with the literature outcomes that illustrate the strong negative effect of nickel ions on the cell viability, metabolic activity, or proliferation of fibroblasts and osteoblasts [90,91,92].

Additionally, the cell adhesion studies indicated quite similar adhesion capabilities of fibroblasts and osteoblasts to adhere to the surface of the Ag-SiO_2_ and HAp coatings (Figure 4b). Similar to other studies, there were no statistical differences between the cells and the coat-forming materials. Moreover, the morphological examination of the cells after 72 h showed a normal cell morphology without any dead or deformed cells, wherein the full spread of most of the living cells indicated a focal phase of adhesion (Figure 4c). This data revealed similar adhesion properties when the hydroxyapatite and silver-silica nanocomposite coatings were compared. Interestingly, the slightly higher adhesion parameters that were found for the Ag-SiO_2_ coating might correlate with the varied morphology of the coat-forming material as was suggested in earlier results [62]. Additionally, those data are slightly lower than the RMS values for the hydroxyapatite- coated plate and illustrate the crucial impact of local surface roughness on the cell morphology and surface adhesion. Another hypothesis to explain this effect could be the slightly higher water contact angle and slightly lower polar part of the SFE that implicates a decrease in the hydrophilicity of Ag-SiO_2_ coatings relative to the reference HAp. These hypotheses may correspond to the literature data that link the cell interactions of the cell with different factors, i.e., surface roughness [93], wettability [94], or surface free energy [95]. Despite the minor differences in the biological studies, silver-silica as a potentially new material for functionalizing the NiTi surface is possible due to its similar biocompatibility and cellular adhesion, especially relative to the reference hydroxyapatite.

## 3. Materials and Methods

The coatings were deposited on a previously passivated NiTi substrate from a colloidal suspension at a concentration of 0.1 wt.% SiO_2_/Ag nanocomposite powder in 50% ethanol using electrophoresis (EPD) (Avantor Performance Materials, Gliwice, Poland). The cataphoretic deposition was performed under a voltage of 5 V and durations from 1 to 15 min. Platinum was used as the counter electrode. Homogenous, continuous, and crack-free coatings were obtained at a voltage of 5 V and a deposition duration of 15 min. Next, the coatings were dried at room temperature for 24 h, and then they were sintered at 700 °C in a technical argon atmosphere for 2 h to sinter the ceramics particles and increase the coating’s adhesion to the metallic NiTi substrate [62]. Reference HAp coating on NiTi alloy were obtained according to procedure previously described in the literature [17,59].

### 3.1. Wettability and Surface Free Energy Measurements

The water contact angles (WCA) of the studied surfaces were measured using a Drop Shape Analyzer DSA-25E goniometer (KRÜSS GmbH, Hamburg, Germany) at (22 ± 2) °C, in a laboratory atmosphere and at (45 ± 5)% relative humidity. Two μL drops were deposited on the surfaces using an automatic syringe. The contact angles were measured three times for each sample at different sites. The photos of the deposited drops were computer-controlled. The water contact angles were measured automatically using Advance software. The surface free energy (SFE) was calculated using the Owens, Wendt, Rabel, and Kaelble (OWRK) method using two liquids: polar (water) and dispersive (diiodomethane) [96]. The data were obtained based on five different measurements per liquid, and each drop was measured three times. The obtained WCA and SFE values are given as the averages with the standard deviation.

### 3.2. Tribological Tests

The friction coefficient and adhesive forces were determined using a ball-on-flat micro tribometer (T-23) equipped with a 5 mm diameter silicon nitride (Si_3_N_4_) ball. The measurements were taken in technical dry friction conditions in a controlled ambient atmosphere. All of the specimens and counterpart movements as well as the load applying mechanism systems were computer-controlled by Lab-View software. The normal loads that were used were in the range of 30–80 mN. The Si_3_N_4_ ball that was used as a counterpart was sliding on the surface of the coating at a velocity of 25 mm/min over a traveling distance of 5 mm. The RMS surface roughness of the counterpart was (5.0 ± 0.5) nm and the hardness was 14.7 GPa. The adhesion of each sample was read from a graph of the relationship between the friction force and load force at the intersection of the straight line with the *x*-axis.

### 3.3. Analysis of the Surface Topography and Roughness Measurements

A Solver P47 Atomic Force Microscope (AFM) (NT-MDT, Moscow, Russia) was used to study the morphology and roughness of the studied surface at the nanoscale. All of the measurements were taken in air under ambient conditions of (22 ± 2) °C and (45 ± 5)% humidity. The topography images were recorded using the tapping mode. The scanned areas were (2 × 2) μm and (40 × 40) μm at a scan rate 0.5 Hz. The root mean square (RMS) roughness parameter was calculated from the AFM images.

### 3.4. Ion Release

The ion release from the samples was investigated during a long-term incubation. The investigated samples were placed separately in falcon tubes each with 20 mL of the Ringer solution (Baxter Healthcare Corporation, Deerfield, IL, USA). The samples were shaken at 60 rpm and incubated at 37 °C for up to 28 days. After 1, 5, 8, 16, and 23 days, the content of titanium, nickel, phosphorus, calcium, silicon, and silver that had been released into the Ringer solution was determined using inductively coupled plasma atomic emission spectrometry (ICP-AES), Varian 710-ES (Santa Clara, CA, USA) equipped with an OneNeb nebulizer. The parameters were as follows: RF power 1.0 kW, plasma flow 15 L/min, auxiliary flow 1.5 L/min, nebulizer pressure 210 kPa, pump rate 15 rpm, emission lines of Ni: λ = 216.555 and 230.299 and 231.604 nm, Ti: λ = 334.941 and 336.122 and 334.188 nm, P: λ = 213.618 nm, Ca: λ = 317.933 and 422.673 nm, Ag: λ = 328.068 and 338.289 nm, Si: λ = 251.432 and 251.611 and 250.690 nm. The calibration curve method was used. The calibration curves were prepared using a matrix (Ringer solution) of the same concentration as the samples. Single-element standard solutions of 1 mg/mL were used. Deionized water was prepared using the Millipore Elix 10 system (Merck Millipore, Darmstadt, Germany). The obtained results were the average of concentrations that were obtained for all of the analytical lines that were used with a standard deviation not exceeding 1.5%.

### 3.5. Immersion in Simulated Body Fluid

The samples were immersed in a simulated body fluid (SBF) with an ion concentration simulating that of human blood plasma. The SBF solution contained these ions at the following concentrations (mol/L): Na^+^ 1.42, K^+^ 0.05, Mg^2+^ 0.015, Ca^2+^ 0.025, Cl^−^ 1.478, HCO^3−^ 0.042, HPO_4_^2−^ 0.010, and SO_4_^2−^ 0.005. The SBF was prepared by dissolving reagent-grade NaCl, NaHCO_3_, KCl, K_2_HPO_4_·3H_2_O, MgCl_2_·6 H_2_O, CaCl_2_, and Na_2_SO_4_ in distilled water, which was then adjusted to a pH of 7.40 using tris(hydroxymethyl) aminomethane and 1M HCl at 36.5 °C. The samples were immersed for up to three weeks, and the solution was replaced every two days. Then, the samples were washed carefully using distilled water, and the surface morphology and the chemical structure were examined using a scanning electron microscope equipped with an energy dispersive X-ray spectrometer (SEM-EDS) (Phenom ProX (ThermoFisher, Eindhoven, The Netherlands); accelerating voltage, 15 kV).

### 3.6. Zone of Inhibition Test

The antimicrobial properties of HAp (as a reference) and Ag-SiO_2_ coatings that had been deposited on the TiO_2_/NiTi alloy were evaluated using the zone of inhibition test. Two bacterial strains, Gram-negative *Escherichia coli* (DSM 1103) and Gram-positive *Staphylococcus aureus* (DSM 799), were used. Briefly, 1 mL of glycerol stock culture (−80 °C) was added to 50 mL Luria Broth (LB) medium and cultured at 28 °C (shaking at 160 rpm) overnight. Obtained bacterial cell suspensions were diluted (1:50) in a fresh LB medium (50 mL) and cultivated (28 °C, 160 rpm) until the mid-exponential growth phase (OD_600nm_ = 0.6) was reached. After the cell suspensions were centrifuged (4700 rpm for 10 min at 20 °C), the pellet was washed in sterile demineralized (DI) water (Milli-Q, Merck Millipore, Darmstadt, Germany) and centrifuged once again. Lastly, the cells were resuspended in DI water to a density of ~10^7^ CFU/mL for the bacteria (OD_600nm_ = 0.1). A 100 µL suspension of *E. coli* or *S.aureus* bacteria was spread on the surface of an LB agar medium in Petri dishes. Surface-modified NiTi alloys that had been coated with HAp or Ag-SiO_2_ were placed on the surface of the medium with microorganisms. The effect of the surface-modified NiTi alloys was evaluated in triplicate. The plates were incubated at 28 °C. The appearance of biofilm that overgrew the edge of NiTi alloy plates or inhibition zone close to the edge of alloy was evaluated after 1, 14 and 28 days. An example of each strain-surface-modified NiTi alloy combination is presented in Figure 2.

### 3.7. Cell Culture and Viability Assay

The human normal fibroblasts (NHDF) and the human osteoblasts (HOB) were purchased from PromoCell (Heidelberg, Germany). The NHDF cell line was cultured in Dulbecco’s Modified Eagle’s Medium (DMEM; Sigma-Aldrich, St. Louis, MO, USA) containing 15% fetal bovine serum (FBS; Sigma-Aldrich, St. Louis, MO, USA), while the HBO was cultured in an osteoblast growth medium (Sigma, Sigma-Aldrich, St. Louis, MO, USA). In both cases, antibiotics were added to the culture medium (1% *v*/*v* of penicillin and streptomycin; Gibco, Grand Island, NY, USA). The cells were grown under standard conditions at 37 °C in a humidified atmosphere at 5% CO_2_. All of the cell lines were tested for mycoplasma contamination using the PCR technique.

A viability assay was performed directly on the studied materials where the cells were seeded at a density of 14,000 cells/cm^2^ and incubated at 37 °C for 72 h. After incubation, the tested materials were transferred to new Petri dishes, and the CellTiter 96 Aqueous One Solution—MTS (Promega, Madison, WA, USA) solution was added to the medium without phenol red. After 1 h, the absorbance of the formazan that had formed, as the indicator of proliferating cells, was measured. The obtained results are expressed as the percentage of the reference hydroxyapatite. Each material was tested in three independent experiments.

### 3.8. Cell Adhesion Assay and Cell Morphology

The cells were seeded at a density of 14,000 cells/cm^2^ directly onto the studied materials and incubated at 37 °C for 72 h. Next, the cells were collected from the surface using trypsin and incubated with Calcine AM (ThermoFisher Scientific, Waltham, MA, USA) for the imaging of living cells at a concentration of 1 µM and Hoechst 33,342 (Sigma, Sigma-Aldrich, St. Louis, MO, USA) to determine the number of cells at a concentration of 5 µM for 30 min in the dark at 37 °C. The cells were then centrifuged (300× *g* for 5 min) and washed gently with the medium without phenol red and serum. The washing procedure was repeated three times. The fluorescence was measured using a multiplate reader (Synergy 4, BioTek, Winooski, VT, USA) at λ_ex_ = 494 nm, λ_em_ = 517 nm for the calcein and λ_ex_ = 345 nm and λ_em_ = 485 nm for the Hoechst 33342. Adhesion was calculated as the percentage of the reference material (hydroxyapatite-coated plates) from three independent experiments.

To examine the morphology of the cells on the surface of the tested materials, the cells were seeded and stained as described above. After staining, the plates were washed three times with the medium without phenol red and fetal bovine serum (FBS). Next, the cells were fixed in 3.5% paraformaldehyde for 1 h and washed three times with the medium without phenol red and FBS. They were observed using Zeiss AxioObserver Z1 equipped with AxioCam RMn camera (Oberkochen, Germany).

### 3.9. Statistical Analysis

The results are expressed as the mean ± standard deviation (SD) from at least three independent experiments. The statistical analysis was performed using the one-way ANOVA with Tukey’s post-hoc test. A *p*-value of 0.05 was considered to be statistically significant. The statistical analysis was performed using GraphPad Prism (version 5) software (GraphPad Software, San Diego, CA, USA).

## 4. Conclusions

The coat-forming silver-silica nanomaterial was tested to determine its potential usefulness as a new class of biomaterial. Here, the surface roughness, wettability, and tribological features were analyzed relative to the reference hydroxyapatite coating. According to the results, the surface roughness ranged from (112 ± 3) nm (Ag-SiO_2_)—(141 ± 5) nm (HAp), the water contact angle ranged between (74.8 ± 1.6)° (Ag-SiO_2_)—(70.6 ± 1.2)° (HAp) while the surface free energy was 45.4–46.8 mJ/m^2^. The adhesive force and friction coefficient were also determined to be 1.04 (Ag-SiO_2_)—1.14 (HAp) and 0.247 ± 0.012 (Ag-SiO_2_) and 0.397 ± 0.034 (HAp), respectively. The chemical data showed that the release of the metal ion from the Ag-SiO_2_ or HAp coating had a negligible effect, wherein all of the coatings had a high biocompatibility. The NiTi alloy that had been coated with Ag-SiO_2_ did not favor the formation of the *E. coli* and *S. aureus* biofilm compared to the HAp-coated alloy, which was protected only by the *S. aureus* formation. The metallic substrate that had been functionalized by the Ag-SiO_2_ and HAp coatings stimulated the normal human dermal fibroblast and osteoblast cell growth and also constituted a kind of scaffold that helped the cells to adhere to the coat-forming material. All of these studies showed that silver-silica seems to be a promising new kind of material for functionalizing the NiTi surface, which can significantly improve bactericidal protection.

## Figures and Tables

**Figure 1 ijms-22-00507-f001:**
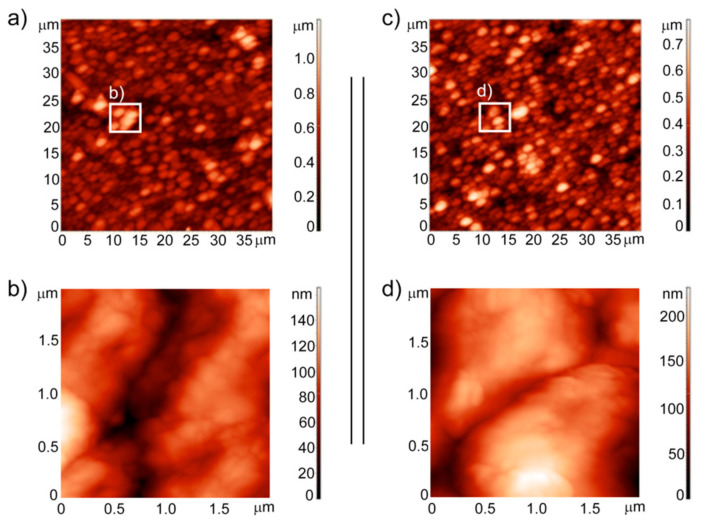
AFM topography of Ag-SiO_2_ measured at (**a**,**c**) (40 × 40) μm and at (**b**,**d**) (2 × 2) μm. The AFM images are visualized for the Ag-SiO_2_ coating before sintering (**left** panels) and after sintering at 700 °C/2 h (**right** panels).

**Figure 2 ijms-22-00507-f002:**
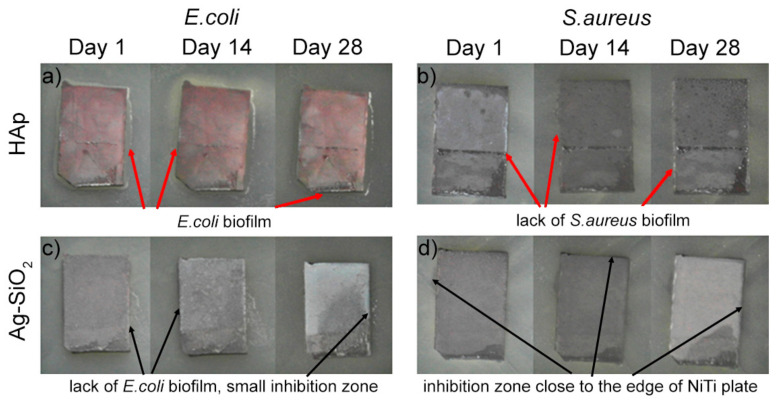
Effect of the (**a,b**) HAp and (**c,d**) Ag-SiO_2_ coatings functionalized the passivated NiTi alloy on the growth of Gram-negative bacteria *Escherichia coli* (**left** panels) and Gram-positive bacteria *Staphylococcus aureus* (**right** panels). The appearance of biofilm that overgrows the edge of NiTi alloys, or its lack of, is red arrow marked. Inhibition zone close to the edge of alloys is black arrow highlighted. The experiment was evaluated after 1, 14 and 28 days.

**Figure 3 ijms-22-00507-f003:**
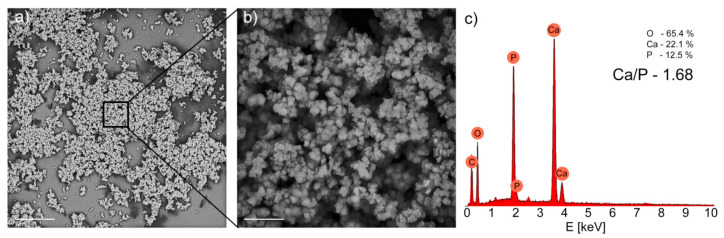
SEM images of the Ag-SiO_2_ that functionalize NiTi surfaces, obtained after a three-week incubation in a simulated body fluid. Scale bars—(**a**) 100 µm and (**b**) 20 µm. (**c**) EDS spectrum with the atomic concentration values for oxygen, calcium, and phosphorous with a Ca/P ratio that was estimated based on the chemical data.

**Figure 4 ijms-22-00507-f004:**
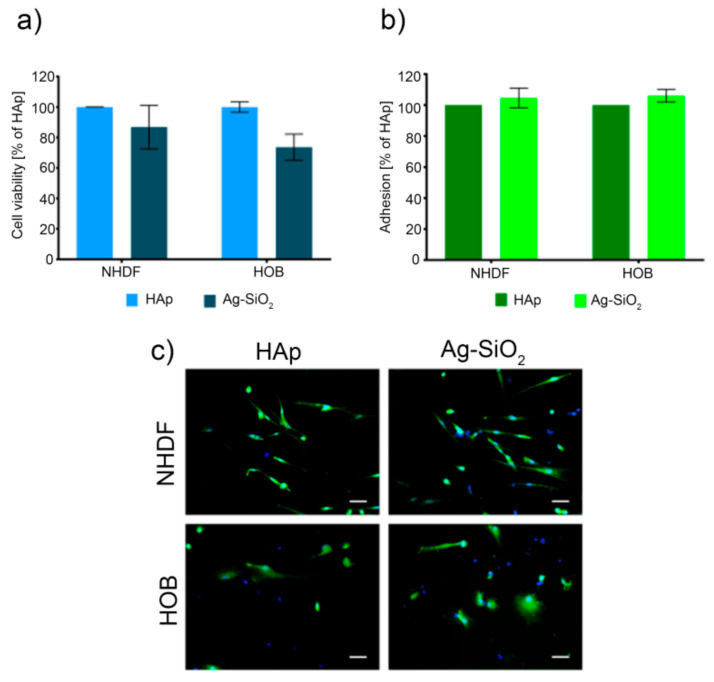
(**a**) Cell viability, (**b**) cell adhesion, and (**c**) cell morphology observed for the hydroxyapatite (HAp) and silver-silica coating (Ag-SiO_2_) that had functionalized the surface of the NiTi alloy. Scale bar—50 µm. Normal human dermal fibroblast (NHDF) and osteoblast (HOB) cells were considered during the studies.

**Table 1 ijms-22-00507-t001:** Friction coefficient (f), adhesive force, water contact angle (WCA), total surface free energy (SFE) with the dispersive (D) and polar (P) components that were obtained for the pristine and sintered HAp and the Ag-SiO_2_ coatings that were deposited on TiO_2_/NiTi. The standard deviation was estimated based on five measurements.

		f	Adhesive Force [mN]	WCA [°]	Surface Free Energy		
					SFE [mN/m]	D [mN/m]	P [mN/m]
HAp	pristine	0.415 ± 0.025	2.00	65.0 ± 1.4	62.2	44.5	17.7
	after sintering	0.397 ± 0.007	1.14	70.6 ± 2.3	46.8	38.5	8.3
Ag-SiO_2_	pristine	0.424 ± 0.003	1.99	65.7 ± 2.1	61.0	47.7	13.3
	after sintering	0.247 ± 0.012	1.04	74.8 ± 1.6	45.4	40.1	5.3

**Table 2 ijms-22-00507-t002:** Results of the ion concentration in the Ringer solution from the NiTi. Statistical analysis was calculated based on five repetitions and was calculated as the standard deviation.

Duration of the Immersion [Days]	Ion Concentration [mg/L]	
	Ni	Ti
1	0.10 ± 0.05	<0.01
5	0.48 ± 0.20	
8	0.66 ± 0.20	
16	1.24 ± 0.77	
23	1.29 ± 0.80	

**Table 3 ijms-22-00507-t003:** Results of the ion concentration in the Ringer solution from the Ag-SiO_2/_TiO_2/_NiTi. Statistical analysis was calculated based on five repetitions and was calculated as the standard deviation.

Duration of the Immersion [Days]	Ion Concentration [mg/L]					
	Ni	Ca	P	Si	Ag	Ti
1	0.05 ± 0.04	90.25 ± 0.49	<0.05	0.03 ± 0.01	<0.01	<0.01
5	0.13 ± 0.05	104.34 ± 0.03		0.08 ± 0.01		
8	0.15 ± 0.04	107.28 ± 0.16		0.11 ± 0.01		
16	0.19 ± 0.03	109.40 ± 0.50		0.19 ± 0.02		
23	0.18 ± 0.03	109.64 ± 0.52		0.20 ± 0.01		

## Data Availability

Data are stored at the cloud and stick in the form of backup.

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
