# Peer review of "Key Properties of a Bioactive Ag-SiO2/TiO2 Coating on NiTi Shape Memory Alloy as Necessary at the Development of a New Class of Biomedical Materials"

_ijms, 2021, doi:10.3390/ijms22020507_

Round 1
Reviewer 1 Report
Please, clearly indicate the novelty of the Ag-SiO2 coating compared to other coatings, for example hydroxyapatite. In particular, the innovative prospect of the future development of the Ag-SiO2 coating compared to the "conventional" ones (such as hydroxyapatite) is not clear. Authors should better highlight the advantages of their coating than others.
Author Response
You have right and we would like to thank you for the valuable comment. In the newest version, we have tried to correct those mistakes and make the paper more clear. As a result, the Introduction has been completely rewritten and new points have been added in the newest version, especially in the context of novelty. We hope that the paper is now much better clarified and our aim had been achieved.
Reviewer 2 Report
The manuscript covers a very interesting topic such as development of biomaterials for clinical applications. A comparison between Ag-SiO2 coating and hydroxyapatite coating of NiTi shape memory alloy is presented. Thus, the chosen interdisciplinary approach represents the goal of the present study: chemical-physical analyses of the material are combined with biological assays. However, in the actual form, the latter is not clear and need major revisions. It is difficult to appreciate the ratio of the biological assays performed both with prokaryotic and eukaryotic cells. Furthermore, there is a strict link between the submitted paper and the other author’s paper under revision: without additional information is not clear to get the right view.
From a biological point of view, along the text several inconsistencies may be hightlighted. Few are herein listed.
Abstract:
Line#37: “NiTi alloy coated by Ag-SiO2 turned out to be nonsensitive to E. coli and S. aureus biofilm formation concerning the HAp coated alloy.” This sentence is not clear. Microorganisms can be defined sensitive to a certain antimicrobial treatment, on the other hand materials are not sensitive to microorganisms.
Introduction
The introduction should guide readers through the background of the topic. The fabrication of new materials with features useful for clinical applications could be the open. On the other hand, the authors begin citing their paper under revision and the novelty of hybrid coatings. This should be the final part of the introduction in order to make clear the aims of the study.
Some comments are reported:
Line# 77: the term” translates” should be changed.
line# 78: which target does represent“ the cited outer membrane”?
Lines#79-81: “On the other hand, improvement of the coat-forming material through the application of the silver nanoparticles or ions at the concentration lower than 2 at. % prevents microbial biofilm formation, improves pathogenic features of the coat-forming material, as well as stimulates cells’ proliferation”.
This sentence is not clear.
Lines#100-105: “A positive correlation between … and biological response”. Which biological features are desired? The increase in biofilm formation is not a good feature. Which type of cell should adhere to coatings? Host cell?
Lines#111-117: references should be added to support this part.
Results and discussion
The organization of this section in paragraphs could improve the readability. The part referring to “biological” items are not clear.
Lines # 253-254: “Antimicrobial outcomes showed a lack of inhibition of the microbial growth on the reference coating composed of hydroxyapatite”. No data have been shown. Which microorganisms have been used? Which antimicrobial assays have been performed? No images, no data have been shown.
Lines #254-256: “Despite this, a long-term effect led to the formation of the Gram-negative (E. coli) bacterial biofilm, while the hydroxyapatite coatings exposed to the Gram-positive bacteria (S. aureus) turned out to not promote biofilm formation”. How biofilm formation has been evaluated? No results have been shown.
Lines#258-261: “Considering a correlation of our data with the literature, we can expect that the presence of phosphates and/or anchoring the hydroxyl groups on the hydroxyapatite surface may facilitate the formation of a water film on the surface that should hinder the direct interaction between the material and the bacterium.” This sentence is not clear. Does water prevent bacterial adhesion? Indeed, bacterial biofilm form on a surface facing a liquid phase.
Lines#2676-269: ”Hence, different behavior of the hydroxyapatite coatings concerning various bacteria strains generates the environmental problem, especially in the context of further in vivo studies.” Which environmental problems? This sentence is not clear.
Line#271. The citation of table 1 is not relevant for the cited bacterial response.
Lines# 271- 276. The microbiological assays are not well described. Which is the aim of aech test and how has been performed? Which control have been included? The description of a “lack of E. coli biofilm” and “inhibition zone”. Images, measurement of inhibition growth halos should be reported to support the biological assays.
Lines# 331- 334. The observed cellular viability decrease can be analysed and commented if statistically significant.
Material and methods
The biological assays should be written in a manner that other researchers could perform the same experiments. The actual form does not meet this aim. Furthermore, it is very important to include the panel of samples used as controls. The biological controls help to interpret the obtained results.
Author Response
The manuscript covers a very interesting topic such as development of biomaterials for clinical applications. A comparison between Ag-SiO2 coating and hydroxyapatite coating of NiTi shape memory alloy is presented. Thus, the chosen interdisciplinary approach represents the goal of the present study: chemical-physical analyses of the material are combined with biological assays. However, in the actual form, the latter is not clear and need major revisions. It is difficult to appreciate the ratio of the biological assays performed both with prokaryotic and eukaryotic cells. Furthermore, there is a strict link between the submitted paper and the other author’s paper under revision: without additional information is not clear to get the right view.
You have right and we would like to thank you for the valuable comments. In the newest version, we have tried to correct all of those mistakes and have made the paper more clear. We hope that the aim had been achieved.
From a biological point of view, along the text several inconsistencies may be hightlighted. Few are herein listed.
Abstract:
Line#37: “NiTi alloy coated by Ag-SiO2 turned out to be nonsensitive to E. coli and S. aureus biofilm formation concerning the HAp coated alloy.” This sentence is not clear. Microorganisms can be defined sensitive to a certain antimicrobial treatment, on the other hand materials are not sensitive to microorganisms.
It has been corrected.
Introduction
The introduction should guide readers through the background of the topic. The fabrication of new materials with features useful for clinical applications could be the open. On the other hand, the authors begin citing their paper under revision and the novelty of hybrid coatings. This should be the final part of the introduction in order to make clear the aims of the study.
You have absolutely right. The Introduction has been rewritten and new points have been added in the newest version. We hope that the theme of the paper is now much better clarified.
Some comments are reported:
Line# 77: the term” translates” should be changed.
It has been corrected.
line# 78: which target does represent“ the cited outer membrane”?
We mean the bacteria cell wall. It has been corrected in the paper.
Lines#79-81: “On the other hand, improvement of the coat-forming material through the application of the silver nanoparticles or ions at the concentration lower than 2 at. % prevents microbial biofilm formation, improves pathogenic features of the coat-forming material, as well as stimulates cells’ proliferation”.
This sentence is not clear.
It has been rewritten and English corrected.
Lines#100-105: “A positive correlation between … and biological response”. Which biological features are desired? The increase in biofilm formation is not a good feature. Which type of cell should adhere to coatings? Host cell?
It has been rewritten, clarified and English corrected in the newest version.
Lines#111-117: references should be added to support this part.
It has been added.
Results and discussion
The organization of this section in paragraphs could improve the readability. The part referring to “biological” items are not clear.
Lines # 253-254: “Antimicrobial outcomes showed a lack of inhibition of the microbial growth on the reference coating composed of hydroxyapatite”. No data have been shown. Which microorganisms have been used? Which antimicrobial assays have been performed? No images, no data have been shown.
Additional figure has been added to the manuscript. The inhibition zone test was detailed described in the materials and method section.
Lines #254-256: “Despite this, a long-term effect led to the formation of the Gram-negative (E. coli) bacterial biofilm, while the hydroxyapatite coatings exposed to the Gram-positive bacteria (S. aureus) turned out to not promote biofilm formation”. How biofilm formation has been evaluated? No results have been shown.
It has been corrected. Additional descriptions were presented in Figure 2.
Lines#258-261: “Considering a correlation of our data with the literature, we can expect that the presence of phosphates and/or anchoring the hydroxyl groups on the hydroxyapatite surface may facilitate the formation of a water film on the surface that should hinder the direct interaction between the material and the bacterium.” This sentence is not clear. Does water prevent bacterial adhesion? Indeed, bacterial biofilm form on a surface facing a liquid phase.
We edited this misleading sentence. English has been corrected.
Lines#267-269: ”Hence, different behavior of the hydroxyapatite coatings concerning various bacteria strains generates the environmental problem, especially in the context of further in vivo studies.” Which environmental problems? This sentence is not clear.
It has been corrected.
Line#271. The citation of table 1 is not relevant for the cited bacterial response.
We have corrected this sentence and connected this reference directly to ‘wettability and roughness parameters’.
Lines# 271- 276. The microbiological assays are not well described. Which is the aim of aech test and how has been performed? Which control have been included? The description of a “lack of E. coli biofilm” and “inhibition zone”. Images, measurement of inhibition growth halos should be reported to support the biological assays.
Additional figure with descriptions has been added to the newest version of the manuscript. The inhibition zone test was described in the materials and method section.
Lines# 331- 334. The observed cellular viability decrease can be analysed and commented if statistically significant.
You have absolutely right. Unfortunately, the biological experiments performed on the hydroxyapatite and silver-silica coatings turned out to be very similar to each other (for us it is fantastic news) with cell viability and adhesion without statistical significance. This information has been discussed in the text and therefore we can not develop this part more. Of course, if the statistical significance had appeared that differentiate the samples we developed this fragment much more, and correlate our data with the literature.
Material and methods
The biological assays should be written in a manner that other researchers could perform the same experiments. The actual form does not meet this aim. Furthermore, it is very important to include the panel of samples used as controls. The biological controls help to interpret the obtained results.
Additional information has been added to the materials and methods section. We evaluated the antibacterial properties of Ag-SiO2 functionalized NiTi alloy and compared the results with HAp-coat-forming material functionalized NiTi alloy (it was our reference). Unfortunately, we did not discuss the uncoated samples. Thank you for valuable suggestion. In our future works, we will compare the data of new coatings with the uncoated reference NiTi alloy.
Round 2
Reviewer 1 Report
The revised version highlights the novelties of the proposed systems better than the original version. Paper can be accepted for publication.
Author Response
We would like to thank the Reviewer for the comment and nice review. We are glad that we have corrected the paper according to your thoughts.
This manuscript is a resubmission of an earlier submission. The following is a list of the peer review reports and author responses from that submission.